# Human Coronavirus Cell Receptors Provide Challenging Therapeutic Targets

**DOI:** 10.3390/vaccines11010174

**Published:** 2023-01-13

**Authors:** Georgina I. López-Cortés, Miryam Palacios-Pérez, Margarita M. Hernández-Aguilar, Hannya F. Veledíaz, Marco V. José

**Affiliations:** 1Facultad de Química, Universidad Nacional Autónoma de México (UNAM), México City C.P. 04510, Mexico; 2Theoretical Biology Group, Instituto de Investigaciones Biomédicas, Universidad Nacional Autónoma de México (UNAM), México City C.P. 04510, Mexico; 3Network of Researchers on the Chemical Evolution of Life (NoRCEL), Leeds LS7 3RB, UK

**Keywords:** coronavirus receptor, ANPEP/ CD13, DPP-IV/ CD26, ACE2, MERS-CoV, SARS-CoV, SARS-CoV-2

## Abstract

Coronaviruses interact with protein or carbohydrate receptors through their spike proteins to infect cells. Even if the known protein receptors for these viruses have no evolutionary relationships, they do share ontological commonalities that the virus might leverage to exacerbate the pathophysiology. ANPEP/CD13, DPP IV/CD26, and ACE2 are the three protein receptors that are known to be exploited by several human coronaviruses. These receptors are moonlighting enzymes involved in several physiological processes such as digestion, metabolism, and blood pressure regulation; moreover, the three proteins are expressed in kidney, intestine, endothelium, and other tissues/cell types. Here, we spot the commonalities between the three enzymes, the physiological functions of the enzymes are outlined, and how blocking either enzyme results in systemic deregulations and multi-organ failures via viral infection or therapeutic interventions is addressed. It can be difficult to pinpoint any coronavirus as the target when creating a medication to fight them, due to the multiple processes that receptors are linked to and their extensive expression.

## 1. Introduction

The designation Coronaviridae refers to a family of viruses that all display protein homotrimers across the whole surface of their viral membrane, giving them a crown-like appearance. The coronaviruses are encapsulated positive-stranded RNA viruses [1], and the family contains four distinct genera (alpha, beta, gamma, and delta coronavirus) all of which encode nearly all functionally identical proteins [2]. Known human coronaviruses belong specifically to the genera *Alphacoronavirus* or *Betacoronavirus*, whose common ancestor infected bats [2]. Table 1 depicts the seven distinct coronaviruses that can infect humans: HCoV- 229E, HCoV- NL63, HCoV- OC43, HCoV- HKU1, SARS-CoV, MERS-CoV, and SARS-CoV-2. The latter three have been the main causes of epidemics in recent years [1,3,4].

The structural glycoprotein known as the spike protein (S) enables the host receptor identification and entry into the cell [12]. The S proteins of coronaviruses must be cleaved to activate the fusion peptide and infect the cell [13]. S proteins have two functional subdomains that allow membrane fusion. The host recognition involves the subdomain 1 (S1), which has a C-terminal domain (CTD) and an N-terminal domain (NTD). The receptor binding domain (RBD) may be in the NTD or the CTD, depending on the virus. The fusion machinery, which consists of the fusion peptide and the heptad repeats required for the fusion, is present in the second subdomain (S2). The S2 sequence is far more conserved among the four genera than the S1 sequence, as the virus must adapt to the receptors of the hosts [14]. The adaptability of the coronaviruses has led to their evolutionary success because they respond rapidly to selective pressures [15,16] and can jump to another species with highly identical receptors [14,17].

Coronaviruses are known to primarily use a host receptor, which can be either a protein or a carbohydrate [17]. It is hypothesized that an ancient coronavirus acquired a host galectin sequence that resulted in the ability to bind carbohydrates, so binding to carbohydrates is a more recent feature [18]. Even though each coronavirus has a primary host receptor, it has been shown that several S proteins can bind to other components of the cell membrane [19,20,21]. Even more, the S protein’s glycosylation can also interact with the lectins of the host [20]. The reported protein receptors employed by mammalian coronaviruses are the murine carcinoembryonic antigen-related cell adhesion molecule 1a (mCEACAM 1a), aminopeptidase N (ANPEP also CD13), dipeptidyl protease IV (DPP- IV also CD26), and angiotensin converting enzyme 2 (ACE2) [17]. Since the first one is receptor for the murine hepatitis virus (MHV) and it does not infect humans, herein, we will discuss the other three enzymes solely.

## 2. Human Protein Receptors for Coronaviruses Are Proteases

The viral receptors ANPEP, DPP- IV, and ACE2 display functional similarities, starting with the fact that they are all proteases, expressed in the membrane as dimers, and able to be shed from the membrane while still being enzymatically active. These proteases are moonlighting enzymes that participate in numerous processes and are expressed in a wide variety of cell types, and they are regarded as receptors due to their participation in signal-transduction pathways. In Figure 1A–E, the three proteases are shown in association with the coronavirus S proteins’ RBD region.

The three proteases process several substrates sequentially in the same pathways. A variety of enzymes, including ACE2 and ANPEP, are involved in the processing of vasoactive peptides of the renin-angiotensin aldosterone system (RAAS). Angiotensin I and II are digested by ACE2, and angiotensin III is subsequently cleaved by ANPEP to create angiotensin IV, which is then degraded by the same protease. This cascade will be explained in detail latter. The chemokine CXCL11 is initially hydrolyzed by DPP- IV and subsequently by ANPEP [22]. Other substrates could be cleaved by various proteases in order to be active, such as enkephalin, which is cleaved by ANPEP and subsequently by CD10 [23]. Additionally, potential substrates might be involved in other processes, such as digestion, sodium regulation, and blood-pressure regulation, where the proteases are co-expressed in epithelial cells of the brush border of the intestine, the renal epithelial cells and endothelial cells.

### 2.1. ANPEP

Initially discovered for its catalytic activity, the aminopeptidase N (ANPEP, CD13, alanyl peptidase, gp150) (EC: 3.4.11.2) is a glycoprotein of approximately 150 kDa. It was later found to be the same protein as CD13, which was a myeloid leukemia marker [24]. Even though various names have been given to it (including MY7, SJ1D1, and L138), aminopeptidase N (ANPEP) and CD13 remain [25,26]. ANPEP is a zinc^2+^ dependent metalloproteinase and catalyzes the hydrolysis of peptide bonds in the N-terminal end of neutral residues. The catalytic mechanism of porcine ANPEP was characterized as follows: Glu406, His383, and His387 chelate the Zn^2+^ ion, turning a water molecule into a catalyst, the proton is then transferred to Glu384, and finally this proton is passed to the nitrogen group of the protein [27]. Due to their high level of similarity, mammalian aminopeptidases have a comparable catalytic mechanism. In addition to this property, different species host various coronaviruses using the homologue receptor, e.g., pig transmissible gastroenteritis virus (TGEV), feline enteric coronavirus (FCoV), and canine enteric coronavirus (CCoV) [28].

Numerous substrates indicate that ANPEP is involved in a variety of physiological activities. Vasopeptides such as angiotensin III (Ang III, angiotensin 2-8), angiotensin IV (Ang IV); neuropeptides such as enkephalins, endorphins, neurokinin A, and nociceptin; hormones such as kallidan and bradykinin, glutathione, thymopentin, and splenopentin; extracellular matrix proteins such as entactin and collagen type IV; and inflammatory mediators such as kinins, kallidin, tufstin, CXCL11, and IL-8; and hemorphins are some of the substrates of human ANPEP [22,29,30].

ANPEP is a type II transmembrane protein, meaning that its amino terminus is cytoplasmic and its carboxyl terminus is extracellular (Figure 1A). Human ANPEP is composed of 967 amino acids, with a single transmembrane junction, a sizable extracellular region, and a small cytoplasmic portion of only nine residues [22]. Carbohydrates add about 40% of weight to the 109.5 kDa of protein, bringing the total weight to 150 kDa [31].

Noteworthily, ANPEP is expressed in the membrane, where it creates persistent non-covalently coupled homodimers [27]. The expression of this peptidase is constitutive in epithelial cells of the intestine, kidney, glomeruli, endothelial cells, fibroblasts, some neurons, meninges, choroid plexus, pineal gland, paraventricular nucleus, pituitary gland, and myeloid cells. ANPEP has also found to be a marker of myeloid cell differentiation. Interestingly, many cancer cells overexpress this protease [22], and in fact ANPEP has been the subject of research as a potential anti-cancer therapeutic drug [32]. This enzyme can be found in the cell membrane, in vesicles and as a soluble protein.

Despite its nine cytoplasmic amino acids and that is a membrane enzyme, it can trigger signaling cascades [33,34,35,36,37], although it is considered as an orphan receptor as the natural ligand is still unknown. Either as an enzyme or as a receptor, ANPEP is involved in a variety of processes, including angiogenesis, peptide breakdown during digestion, migration, cell adhesion, aggregation, and phagocytosis [22,30,37,38,39,40,41]. 

### 2.2. DPP-IV 

The enzyme dipeptidyl peptidase IV (DPP-IV, CD26, adenosine deaminase 2 (ADCP2), (EC: 3.4.14.5)) has a serine peptidase activity that cleaves dipeptides at the amino terminal end [26,42]. Among its substrates are protein 1 type glucagon (GLP-1), gastric inhibitory protein (GIP), and substance P [43,44]. DPP-IV is a type II transmembrane protein with only one transmembrane passage and six cytoplasmic amino acids, but most of the protein (738 amino acids) is extracellular [43]. Two functional domains have been characterized in the extracellular region; one domain is homologous to an α/β hydrolase with serine protease activity and the other is composed by two subdomains, a cysteine-rich region and a highly glycosylated region (Figure 1B). The structure of CD26 is stabilized by a total of five disulphide bridges and nine N-glycosylations. The glycosylation of DPP-IV eases the formation of dimers and the interaction with other proteins [45], and the glycosylation patterns affect its cellular localization without altering its functions [46].

Exopeptidase DPP-IV prefers proline in the second amino acid position, alanine in a lesser degree, and glycine with the least preference when breaking down peptides [46]. Ser630, Asp708, and His740 all contribute to the catalysis [47]. DPP-IV can be released from the cell as an extracellular soluble enzyme by cleavage of its transmembrane region.

A range of leukocyte populations, including T, B, NK, and dendritic cells, express DPP-IV, as well as fibroblast, endothelial cells, epithelial cells of the kidney, liver, lung, small intestine, esophagus, breast, and prostate [48]. In contrast to TNF-α, which inhibits the expression of the protease, cytokine IL-12 causes an overexpression of this enzyme in activated T lymphocytes [49]. 

DPP-IV is important in the activation and maturation of T lymphocytes as it interacts with different proteins involved in the lymphocyte signaling pathway [50,51,52,53,54]. It also interacts with the enzyme adenosine deaminase (ADA) [55], which in turn interacts with adenosine receptor 2 expressed on dendritic cells [50]. Consequently, when these three proteins interact, a bridge between T cells and DCs is created. DPP-IV is associated with cellular adhesion because it can also bind to fibronectin and type 1 collagen in the extracellular matrix, an ability that metastatic cancer cells exploit [42]. 

### 2.3. ACE2

Angiotensin converting enzyme 2 (ACE2, EC: 3.4.17.23) is an 805 amino acid carboxypeptidase involved in the RAAS that regulates blood pressure; this regulation is focused on managing both blood volume and the resistance of the vascular system. Because of their involvement in this crucial physiological process, the most studied substrates of ACE2 are angiotensin I and angiotensin II, which are hydrolyzed to angiotensin 1-9 and angiotensin 1-7, respectively. Angiotensin 1-7 is a vasodilator, inhibits proliferation, and counteracts the vasoconstrictor effects of angiotensin II [56]. Besides this, ACE2 also has other substrates, such as neurotensin, kinetensin, des-Arg- bradykinin, apelins, and casomorphins, among others [57].

ACE2 is a type I transmembrane protein (Figure 1C–E); most of its amino acids are found on the extracellular side, followed by a transmembrane passage, and 43 cytoplasmic amino acids at the carboxyl terminus. Unlike the aforementioned proteases, ACE2 does have canonical signaling motifs; for example, it has a LIR motif (LC3-interacting region), an SH2-binding motif, a PTB, and a PDZ-binding motif [58]. Additionally, residues Tyr781 and Ser783 in the cytoplasmic region are susceptible to phosphorylation [59]. In fact, these motifs interact with integrin β_3_ and clathrin adaptor AP2 µ2 to enhance endocytosis and with the autophagy receptor MAP1LC3 and GABARAP [59]. On the extracellular side, the C-terminal domain of the protease is homologous to collectrin. It has a binding site for zinc^2+^ and Cl^−^ ions; and it has seven glycosylation sites and three disulphide bridges. 

The expression of this protease is also widespread, including endothelial cells, enterocytes, and Leydig and Sertoli cells; in the renal proximal tubule, heart, testicles, and type II pneumocytes; epithelial cells of bronchi, nose, cornea, brain, liver, and gallbladder [60]; and in the basal epidermal layer of the skin [61]. The expression of ACE2 in the epithelial cells of the small intestine is crucial for the co-expression of the neutral amino acid transporter [61]. ACE2 may be found on the cell membranes, in vesicles, secreted, and in the cytoplasm. In fact, the metalloproteinase ADAM17, also known as the tumor necrosis factor-a-converting enzyme (TACE), cleaves the membrane ACE2 and releases it into the interstitial space [62]. In addition, ADAM17 catalyzes the conversion of pro-TNF-α to TNF-α, which can increase ACE2 expression [60]; this upregulation has also been observed in the case of exposure to tobacco smoke and the chronic presence of inflammatory cytokines such as IFN-α, IFN-γ, and IL-13 [63].

The functions that depend on ACE2’s enzymatic activity include the maturation of vasoactive peptides, fluid pressure homeostasis, promoting myocyte contraction, and regulating cell proliferation [64]. The enzymatic site is apart from the binding site from SARS-CoV and SARS-CoV-2, so infection does not perturbate the enzymatic activity. Nonetheless, after either of the two viruses’ entry, ACE2 is downregulated, causing alterations of the RAS. 

## 3. Structural Dissimilarity among the Human Coronaviruses

Interestingly, the RBD of HCoV-NL63 that binds to ACE2 (PDB: 3KBH) and the RBD of HCoV-229E (PDB: 6ATK) that binds to CD13/ANPEP are structurally similar (Figure 2A) and they both cause common cold symptoms in non-immunocompromised people. The aforementioned coronaviruses belong to the *Alphacoronavirus* genera. On the other side, the S proteins of SARS-CoV (PDB: 2AJF) and SARS-CoV-2 (PDB: 6M0J) and HCoV-NL63 (PDB: 3KBH) bind the same ACE2 receptor but the third one in a different fashion (Figure 2B). Finally, all the three pandemic coronaviruses belong to the *Betacoronavirus* genera and bind similarly to their respective receptors (Figure 2C), although MERS-CoV binds to the protease CD26/DDP-IV. 

## 4. Role of Proteases in Physiological Processes

### 4.1. Role of the Proteases in Digestion

Digestion is the conjunction of the sequential process of breaking down food, so that its nutrients can be taken up and absorbed. The digestive system is essential for hormone secretion; it provides the body with amino acids, lipids, carbohydrates, and micronutrients that regulate the immune system and interact with the microbiome, all of which have a variety of effects on the organism. As a result, the digestive system is essential for preserving the homeostasis of the body. Specialized epithelial cells along the gastrointestinal tract express enzymes that catabolize macronutrients and protein transporters to absorb them. Proteins from food are hydrolyzed into oligopeptides and amino acids through the gastrointestinal tract and then absorbed in the small intestine. Small intestinal epithelial cells express the digestive enzymes ACE2, ANPEP, and DPP-IV, which are crucial for the last stages of digestion. 

ACE2 has been described as a key regulator of dietary amino acid homeostasis, antimicrobial peptide expression, maturation of the local immune system, and interaction with the microbiome [61]. Hashimoto and collaborators write that *Ace2* knockout mice have an increased propensity to develop severe colitis and showed that this inflammatory condition was associated with malnutrition. As ACE2 is necessary for the expression of the neutral amino acid transporter B^0^AT1 (*SL6A19* gene), the mice were unable to absorb neutral amino acids from their diet, and consequently serum levels of valine, threonine, tyrosine and tryptophan were reduced [67]. Evidence shows that ACE2 is necessary not only for its functions as a digestive enzyme but also for amino acid absorption, aiding homeostasis [61]; this is also true for ANPEP, which co-localizes with B^0^AT1 on the luminal membrane of the small intestine in rats [68].

### 4.2. Role of the Proteases in Angiogenesis

Angiogenesis is the processes by which new blood vessels develop from existing ones, to supply nutrients and oxygen to all tissues. In mice with arthritis, soluble ANPEP (sANPEP) induced endothelial cell migration and monocyte mobilization [69]. In response to hypoxia, ANPEP and other angiogenic factors are upregulated [70]. ANPEP suppression by interference RNA (RNAi) resulted in the inhibition of capillary tube formation of HUVEC on Matrigel [71]. Similarly, Rinkevich et al. 2015 reported that inhibition of DPP-IV decreases skin-scar formation during wound healing. This occurs because DPP-IV cleaves the angiogenic factor, the high mobility group box 1 protein (HMGB1), at its N-terminal region, which induces endothelial cell migration and capillary-like structure formation, as well as vascular network formation [72]. Moreover, another substrate, the stromal-cell-derived factor alpha (SDF-1a) induced mobilization of bone-marrow-derived stem cells and the expression of the angiogenic vascular endothelial growth factor (VEGF) [73]. The role of ACE2 during angiogenesis lies in the product Ang (1-7), which promotes angiogenesis and cell proliferation via the receptor Mas [74]; besides this, the activation of this receptor in the brain also decreases oxidative stress, neuroinflammation, improves cerebral blood flow and therefore neuronal survival [75], and tissue repair [76,77,78], but also tumor growth [74,79]. That is, that the participation of these enzymes in angiogenesis lies in the signaling generated by their substrates.

### 4.3. Role of the Proteases in the RAAS System

The RAAS is a regulator of the osmolarity of the whole organism: it affects blood ions concentration, blood volume, pressure, and vascular wall resistance; part of the system promotes sodium reabsorption and fluid retention, which increases blood pressure, while the other promotes fluid excretion [60]. The kidneys, lungs, suprarenal glands, the entire vasculature, and the brain have important roles in the system. Although RAAS affects the entire body, there are local RAASs that might primarily affect a specific organ [80]. By sensing blood pressure in the vasculature, this mechanism promotes an increase in blood flow. Under physiological conditions, the RAAS peptides are produced in a dynamic and balanced manner, and any alteration could lead to an acute event of poor blood pressure regulation or a chronic cardiometabolic disorder; in fact, RAAS connects the cardiovascular, renal, and the metabolism systems. 

Figure 3 can help to understand the global RAAS. The juxtaglomerular cells in the kidneys cleave pro-renin in response to a drop in blood pressure and sodium in blood. Following this, renin cleaves angiotensinogen to produce angiotensin I, which in turn is transformed to angiotensin II (Ang II) by the angiotensin converting enzyme (ACE). The two receptors for Ang II, type I (AT1R), and type II receptors (AT2R) cause a variety of effects depending on tissue and receptor type. For example, vasoconstriction is induced via AT1R, which increases sodium reabsorption in the kidney and stimulates the production of aldosterone in the adrenal cortex; in the brain, they promote water intake and retention via the hypothalamus–pituitary gland axis, which releases the antidiuretic hormone [75,81]. The overall consequence is an increase in vasculature tone. AT2R is associated to wound healing and tissue repair, inhibits ACE, and enhances the activity of angiotensin receptor blockers (ARB) [82].

Ang II is regulated by multiple proteases that cleave it to produce different vasoactive peptides that cause a range of effects. The protease (discussed here) implicated in the RAAS with the most evidence in the literature is ACE2, because it hydrolyses angiotensin I (Ang I) producing Ang (1-9), and Ang II producing Ang (1–7) and angiotensin A; subsequent hydrolysis of Ang (1-9) by ACE produces the second metabolite of ACE2. As mentioned above, vasoactive peptides induce vasodilation or vasoconstriction depending on the receptor stimulated. The receptor Mas (MasR), specific for Ang (1-7), promotes anti-inflammatory, antifibrotic, and vasodilatory properties [60]. Different polymorphisms of the *Ace2* gene are known to be associated with RAAS pathologies at different levels [62].

Additionally, aminopeptidases play crucial functions in blood-pressure regulation [81]. In the brain, the inhibition of ANPEP activity in the intracerebroventricular space increases arterial blood pressure [29], as its substrate, Ang III, accumulates, causing blood pressure to rise. In addition, the inhibition of ANPEP increases the release of vasopressin, the antidiuretic hormone that retains water, thus contributing to increased vessel tone [81]. In the renal tubule, ANPEP, via Ang IV, decreases basolateral Na^+^/K^+^ ATPase, thereby increasing urinary Na^+^ excretion and decreasing transcellular Na^+^ transport in the renal artery. Therefore, ANPEP promotes natriuresis by increasing Ang IV and reducing the Ang III concentration.

The precise molecular interactions between DDP-IV and RAAS are still unknown; what is clear is that DDP- IV coimmunoprecipitates with the sodium/hydrogen exchanger-3 in the proximal tubules of the nephrons, so its inhibition reduces the expression of the exchanger and thus the sodium uptake [46]. Besides this, renal accumulation of Ang II induces both glomerular and proximal tubule injury by reducing the expression of megalin, which promotes the reabsorption of filtered albumin and other low-molecular-weight proteins. Nonetheless, Ang II stimulates the activity of DPP-IV, which, together with ANPEP, supports the reabsorption of oligopeptides through its catalytic activity [83].

### 4.4. Role of the Proteases in Metabolism

Metabolic dysregulations, such as obesity and diabetes, cause other complications, such as cardiovascular disease and chronic kidney disease. The role of DPP-IV in metabolism is to degrade the glucagon-like peptide-1 and -2 (GLP1/GLP-2) and glucose-dependent insulinotropic peptide (GIP), which leads to insulin secretion [46]. DPP-IV is upregulated in patients with type 2 diabetes mellitus; this increases insulin resistance and confers cardiovascular risk [46]. Treatment that inhibits DPP-IV activity improves glucose tolerance, hypertension, and has positive consequences in different tissues such as the heart, vasculature, adipose tissue, and kidney [46,72]. Furthermore, obese and insulin-resistant patients have elevated levels of sDPP-IV, which contributes to inflammation by activating T cells via CD45 contacts and activating antigen-presenting cells via interactions with the mannose-6-phosphate receptor and adenosine deaminase [84,85]. 

The role of ACE2 in glucose metabolism is still uncertain and it is thought to act as a compensatory mechanism to reduce the effects of the Ang II/AT1R axis [86]. Elevated Ang II concentrations can lead to hyperglycemia, dyslipidemia, impaired vascular function, and inflammation. ACE2 deficiency increased vascular inflammation and atherosclerosis due to the Ang II/AT1R axis, whereas stimulation of the ACE2-Ang-(1-7)/Mas axis was reported to reduce obesity [84]. 

Increased plasma ACE2 concentration is associated with an increased risk of heart failure incidence, myocardial infarction, stroke, and diabetes [87]. Furthermore, patients with impaired fasting glucose, impaired glucose tolerance, or diabetes had high urinary ACE2 levels as well as high fasting blood glucose and triglyceride, elevated high-sensitivity C-reactive protein, high serum creatinine and urinary albumin-to-creatinine ratio, and elevated systolic blood pressure [88,89]. 

### 4.5. Role of the Proteases in the Respiratory System

Under physiological conditions, the viral receptors are expressed in the epithelial cells of the respiratory airways. ACE2 is co-expressed with TMPRRS2 and cathepsin L in type II pneumocytes, endothelial cells, and the nasal epithelial cells [90]. They must control blood pressure as part of their role as enzymes in the tissue. ACE2 processes Ang II into Ang (1-7), and other vasopeptides that contribute to maintaining the blood pressure in the capillaries surrounding the lungs and upper airways. ANPEP is present in epithelial cells and alveolar macrophages and it could aggravate lung injury by enhancing the generation of reactive oxygen species and NF-kB activation during inflammation [91]. DPP-IV, which is expressed in epithelial cells and T lymphocytes, exhibits comparable behavior. The three enzymes process peptides from the mucosa but have an important role in immunomodulating the tissue [92]. 

## 5. Pathogenesis of the Viruses 

Human coronaviruses infect the organism through cells expressing any of their principal receptors and co-receptors that will process the viral S protein. They enter the body either through the respiratory airways or the gut, primarily affecting the lungs and gastrointestinal tract, but multi-organ failure is imminent. For instance, HCoV-229E, HCoV-NL63, HCoV-OC43, and HCoV-HKU1 cause common colds by infecting superior airways while SARS-CoV, MERS-CoV, and SARS-CoV-2 can also affect class II pneumocytes from the lungs. Coronaviruses have a tissue tropism towards the receptors and the proteases needed for S protein processing [13]. Once the virus has entered the body, it interacts with the matching host receptor using its S protein, and several host proteases fully activate it to allow the membrane fusion [93]. The host’s proteases prime the S protein by cleaving the S1 from S2, allowing the fusion of viral and host membranes. The various proteases cleave at various points during the replicative cycle; the transmembrane serine protease (TMPRSS 2), elastase, plasmin, and trypsin catalyze at the membrane and extracellular levels; the cathepsin-L in endosomes and the furin-like convertases cleave the newly synthesized S protein at the Golgi apparatus [13]. The replicative cycle begins upon the fusing of either the cell membrane or the endosome membrane [94]. 

There are no reports indicating symptoms other than fever, throat pain, dry cough, tiredness, headache, runny nose, and diarrhea for the coronaviruses that cause common cold. However, the pandemic of coronaviruses (SARS-CoV-2, SARS- CoV and MERS-CoV) could cause more complicated symptoms. Due to their entrance path and the presence of the receptors, the most notable and primary pathologies are located in the respiratory airways, yet the gut may suffer tissue damage as well as other organs [95]. Acute myocardial injury, myocarditis, and other cardiopathies, hyperglycemia, diabetic ketoacidosis, gastrointestinal abnormalities, elevated aminotransferases and bilirubin, neurologic distortions, thromboembolic events, and renal injury are some of the complications associated with severe COVID-19 [95]. 

The pathology of the coronavirus infection starts with tissue damage and the loss of function of the infected cell. The immune response and pathophysiology of the common cold coronaviruses remains unclear, but certainly the response is known for pandemic coronaviruses. Coronaviruses infection provokes an immunological signature where proinflammatory cytokines are released (IFN, IL-6, IL-8, IP-10, TNF-α, MCP-1, IL-1β, IL-17, etc.), the accumulation of neutrophils and macrophages in the infected tissue, and lymphopenia of both T and B cells [92]. The infection causes inflammasome NLRP3 activation, which ends in an uncontrolled death driven by pyroptosis and the release of IL-1β [96]. It has been reported that SARS-CoV and MERS-CoV are both detected by the innate receptor toll-like receptor 7 (TLR-7), which triggers the synthesis of pro-inflammatory cytokines, such as type I IFN, IL-6, TNF-α, and the IFN inducible genes [97,98], these communicate to neighbor cells to activate an antiviral state. SARS-CoV-2 is also susceptible to type I IFN [99], in fact we now know that the timing for its synthesis could drive the course of the diseases. A single cell RNA analysis revealed that type I IFN response is not activated in severe patients [100], and another report showed that severe patients had low serum levels of type I and III IFN and high levels of other pro-inflammatory cytokines [101] as well as the viral titers [102]. It remains to be discovered how the immediate response of these patients came about, because it was reported that the delay of IFN cascade caused a rapid viral replication enhancing lung damage and a robust pro-inflammatory cytokine response in a mouse model [103]. 

The consequence of the proinflammatory cytokine production is the recruitment of immunological cells who can combat and repair the tissue. Nevertheless, this may cause edema, impeding lung function and causing poor gas exchange. Liquid in the pulmonary tissue causes cell death, which in turn causes loss of function and increases inflammation. Beside this, it is likely that the capillaries at the lungs worsen the scenario, either by endothelial cell infection or by capillary wall disruption [104]. The latter causes the activation of the coagulation cascades, increasing the possibility of micro thrombi deposition in the pulmonary network, activating systemic inflammatory mediators, and enhancing other coagulopathies [61]. In the case of SARS-CoV-2, the RAAS dysregulates because the binding of the virus to the ACE2 receptors enhances vascular permeability and the disruption of intercellular junctions. All these contributes to lung inflammation, cell death, and fibrosis. 

The catalytic sites of the aforementioned enzymes are not blocked when they interact with the corresponding coronavirus S protein; however, any coronavirus receptor is downregulated because the virus is internalized with it through endocytosis [105]. In fact, DPP-IV and ANPEP are found in membrane domains associated with caveolae [106]. As consequence the product–substrate balance could be altered locally or systemically. Coronaviruses using DPP-IV or ANPEP may have an unbalanced pro-inflammatory response because of their participation in modulating the immune system and cytokine processing. These enzymes are also known to interact with extracellular matrix (ECM) proteins, promote cell migration, adhesion, and tissue remodeling; however, the precise immunological response and the balance of vasoactive peptides must be thoroughly investigated. SARS-CoV-2 pandemics, in contrast to past coronavirus outbreaks, prompted a significant effort to advance understanding of the virus and the physiological reactions of affected organisms. According to studies, SARS-CoV-2 and SARS-CoV entry cause membrane ACE2 downregulation, which puts Ang II and Ang (1-7) out of balance, leading to an increase in inflammation, angiogenesis, pro-oxidation, and pro-fibrotic responses [84]. Protease dysregulation causes change in the concentrations of substrates such as Ang II, apelin-13, dynorphin-13, and products such as Ang (1-7), Ang (1-9), apelin-12, dynorphin-12 [107].

The viruses can infect endothelial cells, causing the exposure of prothrombotic molecules, platelet adhesion, activation of coagulation cascades by the release of plasminogen activator inhibitor 1 (PAI-1), von Willebrand factor (vWF) antigen, soluble thrombomodulin, and tissue factor pathway inhibitor (TFPI), as well as the proangiogenic factors VEGF, hypoxia-inducible factor 1a (HIF-1a), IL-6, and TNF receptor super family 1A and 12 [108]. The vascular damage makes systemic angiopathy and thromboembolism probable, which can lead to multi-organ failure and death [109]. Similarly, myocardium failure is caused by pericytes expressing ACE2 [61]. However, the inhibition of ACE2 catalytic activity could lead to acute myocardial injury associated with SARS-CoV-2 due to Ang II accumulation [62].

### Coronavirus Infection and Co-Morbidities

In the last three years, the world has confronted the confluence of two overlapping pandemics: COVID-19 and metabolic disorders. However, metabolic disorders along with renal disease, cardiac disease, hypertension, lung disease, and any immunosuppressive condition have been reported to be co-morbidities for SARS-CoV, MERS-CoV and SARS-CoV-2 [110,111]. Obesity and diabetes are characterized by chronic low-grade inflammation and oxidative stress, which affect patients in a systemic fashion. In addition, multiple pieces of evidences have shown that COVID-19-induced inflammation exacerbates these inflammatory conditions. The presence of metabolic disorders increases susceptibility to contracting various infectious diseases, but also raises the risk of developing serious complications as a result of inflammation-related damaged tissue [112]. A clinical study showed that patients with type 1 diabetes and type 2 diabetes had 2.86 and 1.8 more risk of COVID-19-related mortality, respectively, in England [113]. The admission to an intensive care unit (ICU) of SARS-CoV patients was 3.1-fold greater if they had diabetes than for non-diabetic patients [114]. 

The dysregulated metabolism in obesity is associated to an imbalance of the RAAS, which in turn promotes the development of cardiovascular and inflammatory diseases [115,116]. Besides this, visceral adipose tissue and resident macrophages of obese individuals express high levels of ACE2, DPP-IV, ANPEP, CD147, and neuropilin-1, as well as furin, which processes S proteins of SARS-CoV-2 [117] and is decreased by anti-inflammatory myokines [118]. COVID-19 severity could be exacerbated through the replication of the virus within adipocytes and through the induction of local and systemic inflammation driven by the infection of adipose-tissue-resident macrophages [119]. Adipose tissue may also act as a virus reservoir [120,121]. Therefore, the tissue might contribute to viral spread with inflammatory responses and cytokine amplification [122]. Additionally, virus infection can promote adipogenesis and increase adipose tissue hypertrophy and hyperplasia [123,124,125]. Apart from diabetes and obesity, furin levels are raised in hypertension [126], which increases susceptibility to severe COVID-19. 

It is important to mention that SARS-CoV-2 can induce diabetes by infecting β pancreatic cells through ACE2, TMPRSS2, and NRP1 [127]. Infection causes cell death but the virus also affects the cell fate of the surrounding surviving cells through a process of trans-differentiation [128]. The final consequence is that insulin production decreases while glucagon increases, raising the blood glucose levels. Patients with previous conditions of uncontrolled high blood glucose levels or dysfunctional β pancreatic cells are at higher risk for developing severe COVID-19.

Obesity and impaired metabolic health generate a potentially reduced immune response, which can negatively affect the immune response [129]. Fortunately, multiple potential molecules are being studied in depth to help patients with metabolic disorders to reestablish their immune system. Tamarind trypsin inhibitor (TTI) shows several beneficial effects on the reduction in inflammatory cytokine TNF-α and biochemical parameters, such as fasting glycemia, triglycerides, and very low-density lipoprotein (VLDL), in addition to improving pancreatic function and mucosal integrity in an obesity model. TTI may inhibit the action of proteases that collaborate with SARS-CoV-2 infection and the neutrophil activity characteristic of lung injury promoted by the virus [130]. TTI may contribute to combating two severe overlapping problems with high cost and complex social implications. 

Kidney injury is one of the fatal consequences of COVID-19, and it is mainly caused by the dysregulation of the RAAS system and the exacerbated concentration of inflammatory mediators and enzymes that leak from the filter system. A previous condition with an altered RAAS or one uncapable of modulating osmolarity renders one more susceptible to coronaviruses infection. However, prophylactic drugs to control blood pressure and other metabolic disorders help to artificially and quickly reestablish physiological conditions. 

## 6. Treatments for Coronavirus Infection

Treatments against coronavirus infections could target different molecules to contain the viral replication. Typically, there are some strategies that attack viral proteins to either prevent the binding of the virus with its cell receptor, intervene in the replicative cycle, or disrupt vesicular transport or protein synthesis. Other treatments target host molecules, aiming to reduce susceptibility to infection, cell damage, repair tissue, inflammation, or other symptoms. Table 2 highlights the treatments that alter viral receptors, which fall in the second group. The alteration of the enzymatic activity and the use of decoy molecules and receptor blockers are some treatments exploited during COVID-19 pandemics [131]; however, similar strategies could be used for other coronaviruses. Proposed treatments range from inhibiting the enzyme or increasing the soluble enzyme to preventing the coronavirus from entering through interaction with the membrane protein. 

Although ANPEP, DPP-IV, and ACE2 catalytic activity inhibitors are tiny molecules used to treat a variety of illnesses, their activity is required for coronavirus infection; thus, they are not the ideal therapy option. The inhibition of the enzyme could worsen the symptoms by accumulating substrates involved in larger physiological processes; for example, inhibiting ACE2 would accumulate Ang II and promote signaling through the AT1R. From this, the blood pressure could raise and worsen the response. Nevertheless, patients with diabetes and obesity may benefit from additional treatment with DPP-IV inhibitors, since they would prevent the inactivation of GLP-1, which would allow the insulin to be released and lower blood glucose levels. By computational modelling, it was observed that three compounds from *Cannabis sativa*, 6-prenylapigenin, cannabivarin, and Δ8-tetrahydrocannabinolic acid-A, resulted in greater affinity for ACE2 than its inhibitor MLN-4760 [136]; further analysis should be undertaken. 

Monoclonal antibodies (mAbs) come from a single clone, so that the protein sequence is the same. Thus, they recognize the same epitope and are therefore considered an efficient therapeutic alternative. The detailed classification of neutralizing monoclonal antibodies that were developed against the S protein of SARS-CoV-2 has been reported [145]. Some of the FDA-approved antibodies for the emergency management of COVID-19 are bamlanivimab/etesevimab, casirivimab/imdevimab, sotrovimab, and bebtelovimab [146]. There have been attempts to identify S proteins of other coronaviruses, including SARS-CoV [147] and MERS-CoV [148]; in fact, an mAb was recently discovered that neutralizes several coronaviruses, including bat and pangolin sarbecoviruses [149]. Even more, pioneering machine learning technology could help reduce investment and increase the number of potential mAbs for a specific purpose, thereby accelerating the discovery of new, more effective therapies of this style [150]. 

Although mAbs is a great strategy for neutralizing viruses, SARS-CoV-2 variants have displayed the rapid mutational rate and the virus is in constant adaptation to evade the antibody recognition of the immune system [15]. Targeting the counterpart by other means, e.g., the host receptors, is not an effective therapy, mainly because the viral receptors are widely expressed in the body and could cause serious problems dysregulating the physiological functions and inducing autoimmunity. An alternative for targeting the receptors is angiotensin converting enzyme inhibitors (ACEI), which reduce the level of Ang II; thus, Ang (1-7) is augmented and signaling through MasR is activated. Additionally, the ACE2 Receptor Blockers (ARB) impede the signaling through AT1R; therefore, Ang (1-7) is produced and accumulates, inducing anti inflammation [151]. The ARB telmisartan was tested in a Phase III clinical trial to moderate COVID-19 patients; it was concluded that there was not an effective reduction in the disease severity, but ongoing trials might assess its effectivity in more severe patients [142]. All these molecules are used for reducing blood pressure because they can modulate the balance between Ang (1-7) and Ang II, favoring the first one; nonetheless, the effectiveness for treating COVID-19 is uncertain [135]. The use of XNT and diminazene aceturate are proposed therapies to enhance ACE2 activity but are still controversial; Haber et al., 2014, demonstrated that the biological effects of these compounds are ACE2-independent [152]. Other experimental reports show that XNT treatment increased the ACE2/ACE ratio and AT1R expression decreased but not AT2R, so both the enzymes and the receptors of Ang II are affected during treatment. Finally, the use of ACE2 as decoy molecules has put different strategies to cheat the virus on the table, with soluble high affinity molecules that could neutralize the virus without altering the physiological processes performed by the receptors [139]. To accomplish this, the soluble receptor or chimeric enzyme should not be enzymatically active, in order to avoid perturbating the RAAS. Further research should be undertaken in order to continue with these promising methods. 

Other drugs that reduce symptoms or evade viral entry were considered for patients with COVID-19. The hosts’ proteases that fully activate the S proteins are interesting targets for impeding the correct internalization, since the viral S protein could not expose the fusion peptide for cell entry. N-0385, is a small-molecule compound that inhibits TMPRSS2 in a nanomolar concentration; it was proven to inhibit SARS-CoV-2 infection in human lung cells [153]. Nafamostat is another TMPRRS2 inhibitor promising to prevent the cleavage of S proteins [154] and the combination of camostat (TMPRSS2 inhibitor) and apilimod (PIK inhibitor) reduced viral entry and prevented viral replication [155]. Other molecules were reported to downregulate TMPRSS2 expression, such as the enzalutamide, an anti-androgen receptor [156]; the homoharringtonine used against chronic myeloid leukemia; and halofuginone, which inhibits T helper cell development [157]. However, drug inhibitors targeting specific entry pathways promote others; for example, the inhibition of TMPRSS2 promotes the omicron variant of SARS-CoV-2 to be dependent on endosome formation and lose the capability to format syncytium [158]. Indeed, the omicron variant is more dependent on cathepsins than the other variants of concern [90], so the use of therapeutic inhibitors of TMPRSS2 will not be as efficient as for other variants. 

The protease cathepsin L is promising as a target because is ubiquitously highly expressed, principally in the respiratory system, gastrointestinal tract, kidney and urinary bladder, sexual tissues, bone marrow and lymphoid tissues, and in endocrine tissues, and its mRNA expression level is higher than angiotensin-converting enzyme 2 (ACE2), FURIN, and TMPRSS2 in human lung tissues [159]. The molecules K777, cathepsin inhibitor 1, E64d, Z-FY-CHO, MDL-28170, and oxocarbazate inhibit cathepsin-L in nanomolar concentrations and prevent its infection to Vero E6 cells [160]. The furin inhibitors decanoyl-RVKR-chloromethylketone and naphthofluorescein also affect it, by impeding the S protein cleavage and therefore syncitium formation; indeed, the effect was reported to be higher than that related to inhibiting TMPRSS2 with camostat. All these molecules could be used against SARS-CoV-2 and against the other coronaviruses because S proteolysis is the one factor required for all the viruses. Hydroxychloroquine and chloroquine are used to treat various viral infections, malaria, and other inflammatory diseases; they increase endosomal pH and interfere with ACE2 glycosylation and cathepsin-L activity [161]. Enzyme inhibitors of all proteases that process the S protein might be promising treatments; however, their wide expression and participation in multiple processes complicate it. There must be effective drug delivery systems in order to effectively impedes virus fusion without affecting other tissues. 

The use of SARS-CoV-2 protease inhibitors is another effective strategy. Two proteases, the 3-chymotrypsin-like protease (3CL^pro^) and the papain-like protease (PL^pro^), contribute to process the new synthetized polyprotein into nonstructural and structural proteins [162]. Given that both proteases are necessary for viral replication, they make potentially intriguing therapeutic targets [130]. Nirmatrelvir, Molnupiravir, Remdesivir, and Paxlovid are inhibitors of the SARS-CoV-2 protease 3CL^pro^, which binds to the catalytic site of 3CL^pro^ [159,163]. Nevertheless, the disadvantages of treating patients with antiviral drugs or mAbs against viral proteins mean that one should constantly check their effectiveness, and because of the high mutation rates of RNA viruses, this could represent a high cost of production. 

Besides this, the use of anti-inflammatory drugs such as corticosteroids and immunomodulating therapies are necessary to reduce the hyperinflammatory state of patients with moderate to severe disease. The Brazilian green propolis EPP-AF was used in a clinical trial to treat hospitalized COVID-19 patients, and they reported a reduction in the length of hospital stay [164]. Other conventional anti-inflammatory drugs have been used, such as dexamethasone or anti IL-6 antibodies, and are possible treatments to reduce the symptoms caused by coronavirus infection. 

## 7. Conclusions

Metabolic disorders, hypertension, and cardiovascular and renal diseases are known to be risk factors for developing severe COVID-19 disease [165], but they are also co-morbidities for all coronavirus diseases. The coronavirus receptors (ACE2, DPP-IV, and ANPEP) are interconnected in physiological processes that impact locally and, more significantly, systemically. Any of these receptors may be involved in chronic conditions that affect vital aspects of homeostasis, such as metabolism, blood pressure, and vascular integrity. ACE2, DPP-IV, and ANPEP proteases are also widely expressed, with a significant role in endothelial cells, the network that connects all tissues and organs. Additionally, several polymorphisms of the proteases, affect the susceptibility and course of coronavirus diseases. Each of these elements has effects on systemic physiology; thus, coronavirus infections can cause multiple organ failure, making it difficult to develop treatments or bullets to block them all. 

Proposed and applied treatments against SARS-CoV-2 has opened a debate regarding the expression of ACE2. On one hand, enhanced ACE2 expression might facilitate the infection and replication of the virus but would favor Ang 1-7 accumulation, and as consequence vasodilation and anti-inflammatory responses. In the other hand, limiting the expression of ACE2 provokes Ang II accumulation, and therefore vasoconstriction and inflammation via AT1R signaling, fibrosis, and tissue injuries. ANPEP and DPP-IV are also involved in processing peptides of the RAAS and metabolism. Therefore, the accurate expression and enzymatic activity of the protein receptors is essential for the maintenance of the homeostasis of the organism. Any alteration could provoke systemic complications, including organ failure. Because of this, diseases that employ these receptors generally create difficulties, especially if they are new viruses that the immune system is not familiar with. The most important issue is how to combat the viral spread and develop prophylactic and treatments against the agents. This represents an important challenge because viral receptors-based drugs represent a double-edged sword against coronavirus infection. 

Although several approaches have been tried, it is obvious that medicine must be more individually tailored to treat each case according to its receptor expression levels and metabolism. Treatment of coronavirus infections with drugs that inhibit the receptor’s catalytic function may result in complex unintended side outcomes. Other treatments that involve modifying the RAAS system have been assessed in clinical trial but findings are inconclusive. The soluble receptors systems may be an attractive alternative for hijacking the virus before it infects the cell. Ongoing research in this field should continue because it is still in experimental phases. Meanwhile, it seems that preventive vaccinations and mAbs against viral proteins can possibly block viral entry without affecting the enzymes’ activity, and a healthy balance of metabolites remains the most effective treatment for coronavirus infections without presenting significant adverse events. Hopefully, the research in soluble molecules systems creates a drug-design platform to solve future outbreaks. 

## Figures and Tables

**Figure 1 vaccines-11-00174-f001:**
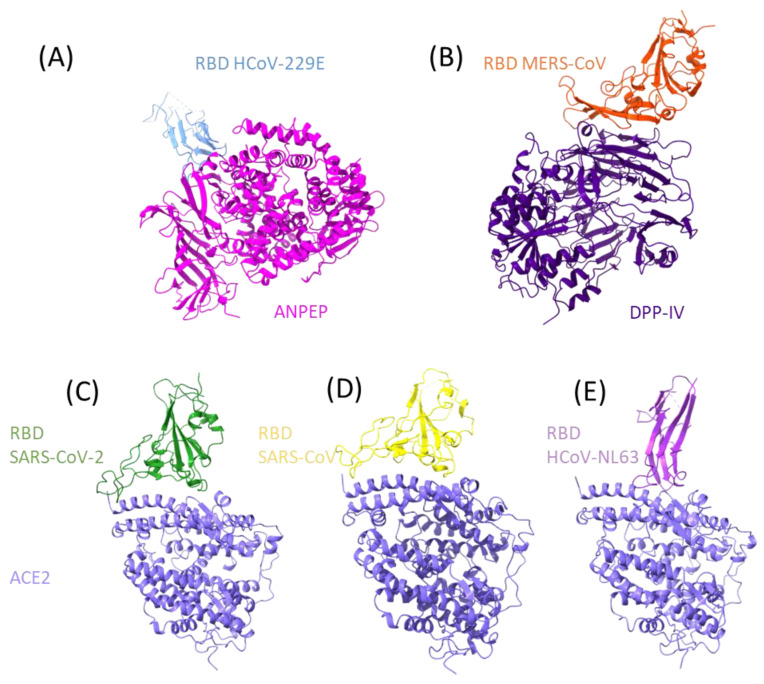
RBD interactions of human coronaviruses with their human protein receptor. The structures of protein–protein interaction of human coronaviruses to the human receptor is presented. At the top from left to right the RBD of HCoV-229E ((**A**), PDB: 6ATK) with ANPEP and the RBD of MERS-CoV ((**B**), PDB: 4KR0) with DPP- IV, at the bottom the RBD of SARS2-S ((**C**), PDB: 6M0J), SARS-S ((**D**), PDB: 2AJF) and HCoV-NL63 ((**E**), PDB: 3KBH) interacting with ACE2.

**Figure 2 vaccines-11-00174-f002:**
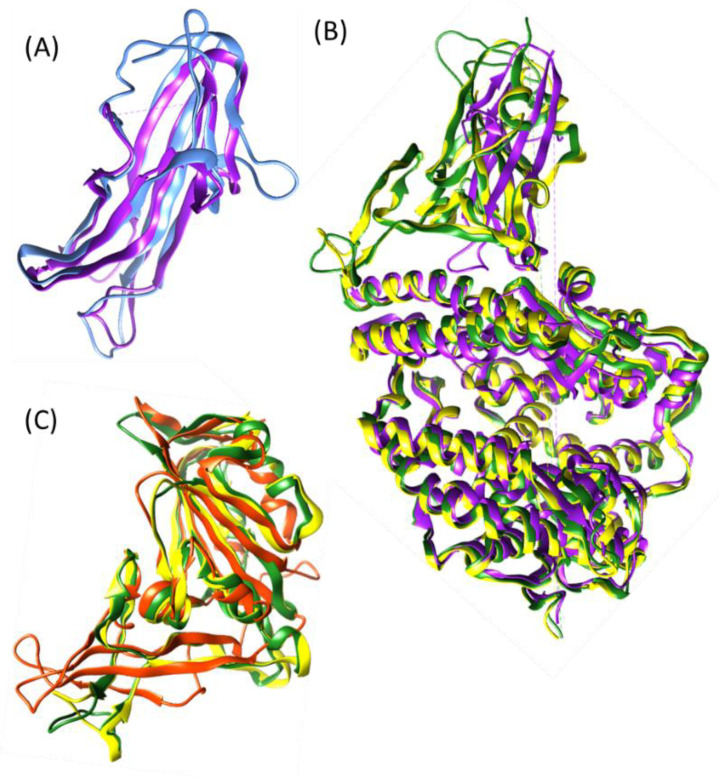
Structural alignments among some viral RBDs of human coronavirus using the US-align protocol [65] and the TM-align tool of the Zhang Lab suite [66]; (**A**): RBD of HCoV-NL63 (PDB: 3KBH,) with the RBD of HCoV-229E (PDB: 6ATK, in blue) alignment of 105 residues with TM-score = 0.83139 and RMSD 1.73; (**B**): RBD of SARS-CoV (PDB: 2AJF, in yellow) and SARS-CoV-2 (PDB: 6M0J, in green) bind the same ACE2 receptor as HCoV-NL63 (PDB: 3KBH, in purple); (**C**): Alignment from the three pandemic coronaviruses RBD (SARS-CoV, in yellow, SARS-CoV-2 in green and MERS-CoV RBD in orange) with TM-score = 0.7 in average, RMSD = 2.73 and 166 residues aligned among the three structures (154 residues aligned between Cov1 and MERS-CoV with TM-score = 0.69194, and 172 residues aligned between CoV-2 and MERS-CoV with TM-score = 0.70488).

**Figure 3 vaccines-11-00174-f003:**
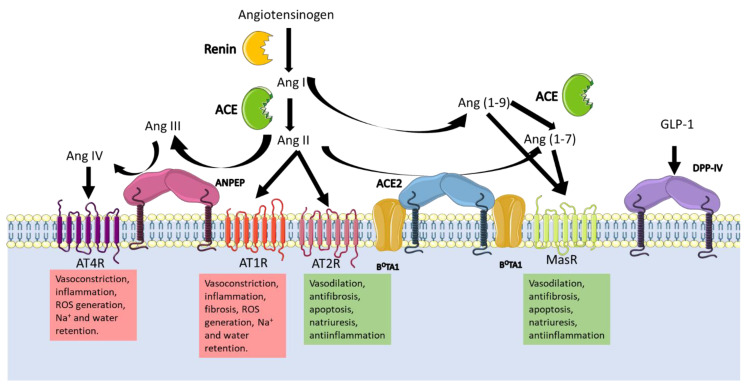
Schematic representation of the peptide flux involved in the RAAS highlighting the participation of the viral receptors ANPEP, DPP-IV, and ACE2. When hypotension is detected, the Angiotensinogen produced in the liver, is catalyzed by the enzyme renin synthetized in the juxtaglomerular cells in the kidney. The product Ang I is either cleaved by the ACE to produce Ang II, which activates any of the Ang II receptors (AT1R and AT2R). Ang I and Ang II can be also cleaved by ACE2 to produce Ang (1-9) and Ang (1-7), respectively. The additional cleavage of ACE to Ang (1-9) produces Ang (1-7) which finally activates the MasR. Additionally, Ang II can be further catalyzed twice by ANPEP producing Ang IV, which activates the AT4R. The effects every Ang II Receptor has, counteracts others so the accurate balance is necessary in order to control blood pressure, osmolarity and inflammation. Ang II is also necessary for aldosterone secretion, which reabsorbs sodium in the kidney, and the antidiuretic hormone release which retains water. The overall effect of the system is the increase of blood pressure. The blockage of the enzymes either with coronaviruses or with other molecules, alters the homeostasis of blood pressure and osmolarity.

**Table 1 vaccines-11-00174-t001:** Human coronaviruses and their most probable ancestor host. Seven human coronaviruses have emerged from zoonotic transmission events.

Coronavirus	Genera	Identified	Most Probable Ancestor Host	Receptor	Ref
HCoV-229 E	*Alphacoronavirus*	1965	Bats *Hipposideros* and camelids	ANPEP/CD13	[1,5,6]
HCoV-OC43	*Betacoronavirus*	1967	Rodents and swine	Sialic acid	[7]
SARS-CoV	*Betacoronavirus*	2002	Bat *Rhinolophus* and civet	ACE2	[8]
HCoV-NL63	*Alphacoronavirus*	2004	Bats *Triaenops*	ACE2	[6]
HCoV-HKU1	*Betacoronavirus*	2005	Rodents	Sialic acid	[8,9]
MERS-CoV	*Betacoronavirus*	2012	Bat and camel	DPP IV/CD26	[10]
SARS-CoV-2	*Betacoronavirus*	2019	Bat *Rhinolophus affini* and	ACE2	[11]

**Table 2 vaccines-11-00174-t002:** Potential treatments for coronavirus infections that target the coronavirus receptors.

Drug Family	Drug	Action Mechanism	Coronavirus Treatment	Other Uses	Ref
ANPEP inhibitors	Ubenimex	Inhibit the catalytic activity of ANPEP.	NR	Cancer treatment in study.	[132]
DPP-IV inhibitors	Gliptins	Inhibit the catalytic activity of membrane and soluble DPP-IV. Reduce glucose levels. Suppress T cell proliferation and pro-inflammatory cytokine synthesis.	Experimental models	Diabetes type 2 treatment. Anti-inflammatory drug.	[84]
ACE2 inhibitors	MLN-4760	Inhibit catalytic activity of ACE2.	Experimental models	Treatment for hypertension, cardiovascular diseases, chronic kidney disease, and diabetes mellitus.	[133]
ACE2 internalization inhibitor	Arbidol	Interacts with aromatic residues within the viral proteins and the plasma membrane. Suppresses the expression of IL-1β, IL-6, IL-12, and TNF-α.	Experimental models	Treatment for various virus including influenza, Ebola virus and hepatitis C virus.	[133,134]
ACE2 viral-binding-site blockers	NAAE (N-[2-aminoethyl]-1 aziridine ethanamine), 6-Prenylapigenin, cannabivarin and Δ8-tetrahydrocannabinolic acid-A	Block the viral docking sites of ACE2 and thus the membrane fusion with the cell membranes.	Experimental and computational models	NR	[133,135,136]
ACE2 as decoy molecules	sACE2 (GSK2586881, APN01, dimeric ACE2)	Binds to extracellular viral S proteins, thus neutralizing the virus. Ang II decreases and Ang (1–7) increases.	GSK2586881 and APN01 in Phase II clinical trial	NR	[137,138,139]
	Chimeric sACE2- IgG Fc fragment	Chimeric protein N terminus of ACE2 with a human IgG Fc fragment at the C-terminus which enhances phagocytosis and complement activation via interaction with Fc receptors.	Experimental models	NR	[140]
Decrease transmembrane ACE2	PMA (phorbol 12-myristate 13-acetate)	Enhances ADAM17 activity to increase ACE2 shedding.	In vitro study	NR	[133]
	Ionomycin	Enhances ADAM10 activity to increase ACE2 shedding.	In vitro study	NR	[133]
	Resveratrol	Reduce ACE2 expression	Phase I clinical trial	Antioxidant, anticoagulant, anti-inflammatory drug	[141]
Increase ACE2 activity	XNT (1-[(2-dimethylamino) ethylamino]-4-(hydroxymethyl)-7-[(4-methylphenyl) sulfonyl oxy]-9H-xanthen-9-one)	Increase ACE2 activity	Experimental model	NR	[133,139]
	Diminazene aceturate	Increases ACE2 activity.	Experimental model	Anti-protozoa drugs.	[133]
	Olmesartan, losartan, telmisartan, azilsartan	ARB increases the ACE2 expression levels.	Phase III clinical trial	Used to downregulate high blood pressure.	[135,142]
	Spironolactone	Increases ACE2 mRNA in macrophages.	Clinical trials not concluded	Treatment for hyperaldosteronism and diuretic drug.	[135,143]
	Apelin-13	Substrate of ACE2.	As hypothesis	Peptide used to treat cardiovascular diseases.	[135,144]

## Data Availability

No new data was generated.

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
