# Peer review of "Human Coronavirus Cell Receptors Provide Challenging Therapeutic Targets"

_vaccines, 2023, doi:10.3390/vaccines11010174_

Round 1

Reviewer 1 Report

The focus of this review are cellular receptors that are utilized by coronaviruses to facilitate entry into target cells, namely the murine carcinoembryonic antigen-related cell adhesion molecule 1a (mCEACAM 1a), aminopeptidase N (ANPEP), dipeptidyl protease IV (DPP-IV), and angiotensin converting enzyme 2 (ACE2). They review the structures and functions of these proteins and the receptor binding domains of different coronaviruses. They also review the functions these receptors play in different tissues and physiological processes, positing that drug therapies aimed at blocking these receptors may also interfere with the functions of these proteins. They then review the pathogenesis of coronaviruses and therapies. 

Overall this is an interesting angle to take on a very extensively reviewed field. The writing is clear but there are English errors throughout and thus further proof reading is needed.

In section 4, there is no discussion of the respiratory system, which seems to be a glaring omission considering its importance in infection and disease pathogenesis.

In section 5, it is not clear that coronaviruses cause respiratory disease. This should be amended and expanded.

I found the section of treatments to be not extensive enough. There needs to be more discussion about each treatment from the point of view of efficacy and side effects. In particular, the authors should discuss if any of these drugs have led to any complications that they suggest they may cause. As this seems to be an important point of the review, yet they do not discuss it at all. 

Table 2 should have information about the stages of development of each drug and include those recommended for use. 

I suggest including the authors view in the conclusion. If these drugs that interact with viral receptors are not optimal, what treatment strategies do they recommend pursuing?

Author Response

In section 4, there is no discussion of the respiratory system, which seems to be a glaring omission considering its importance in infection and disease pathogenesis.

We appreciate this observation, since with good reason we had to be more explicit in the presence of the enzymes in the respiratory system to be able to justify the tropism of the viruses to infect this tissue. Now, we added a subsection mentioning the role of the proteases in the respiratory system.

In section 5, it is not clear that coronaviruses cause respiratory disease. This should be amended and expanded.

In accordance to this observation, we explain in greater detail the pathology of the diseases caused by those viruses, emphasizing the effects in the respiratory system. This was added in section 5.

I found the section of treatments to be not extensive enough. There needs to be more discussion about each treatment from the point of view of efficacy and side effects. In particular, the authors should discuss if any of these drugs have led to any complications that they suggest they may cause. As this seems to be an important point of the review, yet they do not discuss it at all. 

This recommendation helped the article to have more impact and more discussion. We expanded this section to discuss the treatments towards the viral receptors mentioned in Table 2, but we also mentioned other treatments that target different molecules. We compared the use of antiviral drugs and monoclonal antibodies that target the different components of the virus, with the use of drugs targeting host’s proteins that won’t mutate at all. Both strategies should still go under research to develop a better treatment that won’t cause dysregulation in the RAAS and other parameters in metabolism. See section 6.

Table 2 should have information about the stages of development of each drug and include those recommended for use. 

We now include a column indicating whether they are at a clinical phase or if the research is still in experimental models.

I suggest including the authors view in the conclusion. If these drugs that interact with viral receptors are not optimal, what treatment strategies do they recommend pursuing?

We exposed better the conclusion about the difficulties of targeting the coronavirus receptors, if they are involved in various physiological processes. Finally, we draw our position regarding the treatments that are more effective at this moment but indicating the direction where the pharmacological research should go. See conclusion.

Reviewer 2 Report

Congratulations to the authors on a very well written review article. A detailed summary of the key enzymes hijacked by the virus to evade the immune system and cause disease pathogenesis was needed in the field. I have few minor comment s:

- The role of the Furin. TMPRSS2 host proteases in virus S protein activation needs to more clearly summarized in section on Section 6 (treatments) and in section 4 (physiological functions) . There should be greater discussion on possibly targeting these proteins (see Jean et al Nature 2022 etc)

- please discuss Cathepsin L function as well

- there needs to a sub-section that talks about mAbs targeting interactions of viral s proteins with ACE2 , ANPEP in the text under section 6. This angle is need to show how we can block viral entry without possibly affecting ACE2 activity - there is substantial literature on this but key papers need to be summarized here. 

Author Response

Congratulations to the authors on a very well written review article. A detailed summary of the key enzymes hijacked by the virus to evade the immune system and cause disease pathogenesis was needed in the field. I have few minor comments:
Thanks for this comment.
- The role of the Furin. TMPRSS2 host proteases in virus S protein activation needs to more clearly summarized in section on Section 6 (treatments) and in section 4 (physiological functions). There should be greater discussion on possibly targeting these proteins (see Jean et al Nature 2022 etc) -
please discuss Cathepsin L function as well
We were trying to point out the therapeutic developments targeting just the coronavirus receptors as the main players of the article, however thanks to your recommendation we summarized their participation in the pathogenesis of the virus (see section 5) and then we discussed the use of different inhibitors as therapeutics alternatives. See section 6. We appreciate the observation.
- there needs to a sub-section that talks about mAbs targeting interactions of viral s proteins with ACE2 , ANPEP in the text under section 6. This angle is needed to show how we can block viral entry without possibly affecting ACE2 activity - there is substantial literature on this but key papers
need to be summarized here.
We added some examples of mAbs that have been develop to target the viral proteins in order to neutralize the virus. See section 6. In the conclusion section we highlighted the importance of not modifying the enzymes’ activity, property granted by mAbs.
All in all, we thank the reviewers for their time and effort of profoundly reading our manuscript and giving comments and observations that clearly help us improve the article.

Round 2

Reviewer 1 Report

The authors have addressed the comments and concerns raised.